# Physicochemical Properties and Antioxidant Activity of CRISPR/Cas9-Edited Tomato *SGR1* Knockout (KO) Line

**DOI:** 10.3390/ijms25105111

**Published:** 2024-05-08

**Authors:** Jin Young Kim, Dong Hyun Kim, Me-Sun Kim, Yu Jin Jung, Kwon Kyoo Kang

**Affiliations:** 1Division of Horticultural Biotechnology, Hankyong National University, Anseong 17579, Republic of Korea; ajswl1202@naver.com (J.Y.K.); kimdong7916@naver.com (D.H.K.); 2Department of Crop Science, College of Agriculture and Life & Environment Sciences, Chungbuk National University, Cheongju 28644, Republic of Korea; kimms0121@chungbuk.ac.kr; 3Institute of Genetic Engineering, Hankyong National University, Anseong 17579, Republic of Korea

**Keywords:** *sgr1* null lines, lycopene, *β*-carotene, flavonoids, vitamin C

## Abstract

Tomatoes contain many secondary metabolites such as *β*-carotene, lycopene, phenols, flavonoids, and vitamin C, which are responsible for antioxidant activity. *SlSGR1* encodes a STAY-GREEN protein that plays a critical role in the regulation of chlorophyll degradation in tomato leaves and fruits. Therefore, the present study was conducted to evaluate the *sgr1* null lines based on their physicochemical characteristics, the content of secondary metabolites, and the *γ*-Aminobutyric acid (GABA) content. The total soluble solids (TSS), titrated acidity (TA), and brix acid ratio (BAR) of the *sgr1* null lines were higher than those of the wild type(WT). Additionally, the *sgr1* null lines accumulated higher levels of flavor-inducing ascorbic acid and total carotenoids compared to WT. Also, the total phenolic content, total flavonoids, GABA content, and 2,2-diphenyl-1-picrylhydrazyl (DPPH) radical content of the *sgr1* null lines were higher than those of the WT. Therefore, these studies suggest that the knockout of the *SGR1* gene by the CRISPR/Cas9 system can improve various functional compounds in tomato fruit, thereby satisfying the antioxidant properties required by consumers.

## 1. Introduction

Tomatoes (*Solanum lycopersicum* L.) are the most widely consumed major fruit vegetables in the world and are highly available and valuable [1]. Tomatoes contain substances such as sugars, acids, vitamins, minerals, lycopene, and other carotenoids, which contribute significantly to human health maintenance and nutrition [2,3]. Red tomato fruits are used as antioxidants or ROS scavengers because they contain large amounts of vitamins C and E, lycopene, phenol, flavonoids, *β*-carotene, and GABA [4,5]. Tomatoes, with their excellent nutritional value, are consumed in many processed products worldwide. With the daily consumption of tomatoes, the risk of cancer and cardiovascular disease is significantly reduced because the activity of the antioxidants present in them increase in vivo [6]. Carotenoids are molecules of the isoprenoid family that are common to all photosynthetic organs. They are involved in light reception in the photosynthetic membrane of the chloroplast and protect the photosynthetic mechanism from excessive light energy by removing triplet chlorophyll and superoxide anion radicals [6]. Colored carotenoids existing in red, yellow, or orange colors in plants are expressed in flowers, fruits, and roots, and their roles oversee attracting agents for pollination and dispersing seeds. Among them, lycopene, a red pigment, accounts for more than 80% of the total carotenoids in ripe red tomatoes [7]. Lycopene, a fat-soluble compound, has absorption properties like those of dietary fat and is absorbed in the body from the stomach and duodenum. In addition, lycopene absorbed into the body is a powerful antioxidant that removes active oxygen in the body to prevent aging and protect cells to reduce the risk of chronic diseases. Lycopene is synthesized through the carotenoid metabolic pathway and accumulates in the flesh of tomatoes as they ripen, giving them a red color [7,8]. As the tomato fruit ripens, chloroplasts develop into colored bodies, so chlorophyll is lost, and carotenoids accumulate. Previous studies have reported that point mutations in the gene encoding the tomato STAY-GREEN (SGR) protein inhibit chlorophyll degradation during fruit ripening, resulting in an increased chlorophyll and carotenoid content, resulting in brown fruits [9]. In addition, a transformant in which the *SGR1* gene was silenced by RNA interference in tomato had a similar phenotype to that of the *green flesh* mutant [10]. Therefore, *SGR1* plays a crucial role in chlorophyll degradation [11]. In addition, it has been reported that the *SlSGR1* gene, which encodes the tomato STAY-GREEN protein, regulates lycopene accumulation through interaction with the *SlPSY1* gene, a major carotenoid biosynthetic pathway enzyme [12]. Recently, the knockout of *SlSGR1* through the CRISPR/Cas9 system has been found to affect the process of chlorophyll degradation and carotenoid biosynthesis, resulting in color changes in tomato fruits and greatly altering the expression levels of many genes related to photosynthesis and chloroplast function [10,13]. In addition, studies on lycopene accumulation have been performed in model plants such as *Arabidopsis*, rice, red pepper, and soybean [14,15,16,17].

We previously generated *slsgr1* knockout transformants using the CRISPR/Cas9 system to understand the expression of the *SlSGR1* gene during tomato fruit development stages. Then, null plants from which T-DNA was removed were selected and named the *sgr1* #1-6 and *sgr1* #2-4 lines. Unlike the WT, which is a pink tomato, these null lines showed a reddish-brown color in the flesh located under the tomato skin. Compared to the WT, the levels of lycopene and *β*-carotene in the selected lines showed differences depending on the fruit development stage but showed a large difference at the ripening stage. In this study, the antioxidant and physicochemical characteristics of these null lines were investigated to increase consumer acceptance of tomato fruits. To this end, physical characteristics such as firmness, color, TSS, TA, and BAR were investigated together.

## 2. Results

### 2.1. Expression Pattern of the SGR1 Gene

To investigate the expression pattern of the *SGR1* gene, qRT-PCR was performed by isolating mRNA from the tissues of the tomatoes at each development stage. The expression of the *SGR1* gene in the roots, stems, and young leaves showed low levels. However, the expression of the *SGR1* gene according to the development stage of the mature leaves and fruits was high, and it was highest in the Br+7 stage (Figure 1).

### 2.2. Molecular Characteristics of sgr1 Null Lines

To investigate the potential roles of *SGR1* in tomato fruit, an *sgr1* null line was produced using CRISPR/Cas9 technology by targeting the third exon and the fourth exon (Appendix A). A frameshift mutation with a 19 bp (CATGTCATTGCCACATTA) and 5 bp (TGTTA) deletion in the *sgr1* null lines was obtained by Sanger sequencing (Appendix A). The comparison of the predicted amino acid sequences demonstrated that the 19 bp and 5 bp deletions in the *sgr1* gene resulted in the production of truncated protein due to a frameshift and the occurrence of a premature stop codon (Figure 2A). Western blotting analyses also demonstrated that the *SGR1* protein completely vanished in the tomato tissues (Figure 2B). In addition, a PCR analysis was performed using four primer sets present between LB and RB, respectively, to investigate whether T-DNA was free (Figure 3, Appendix A). As a result, no amplification product was observed in the selected null line.

### 2.3. Physiochemical Properties of sgr1 Null Lines

The *sgr1* null lines were subjected to two self-fertilization processes to harvest T4 seeds. The average fruit weight of the *sgr1* null lines ranged from 220 to 230 g. In the red aging phase (fully red reaching, Br+10), the hardness of the *sgr1* #1-6 and *sgr1* #2-4 lines was 2.03 to 2.13 N (Table 1), which was slightly lower than that of the WT (2.80 N). The TSS values of the *sgr1* #1-6 and *sgr1* #2-4 lines at the red ripe stage were in the range of 5.71-5.79 °Brix, which was higher compared to those of the WT, with values ranging around 4.70 °Brix. The mean total acid content of the skinned pulp of the *sgr1* #1-6 and *sgr1* #2-4 lines ranged from 0.41 to 0.42 mg CAE (citric acid equivalent)/10 g dry weight, which was like that of the WT. The average BAR of the *sgr1* #1-6 and *sgr1* #2-4 lines ranged from 13.60 to 14.10, higher than that of the WT.

### 2.4. Colorimetric Evaluation

The Hunter values and antioxidant components in the tomato pulp of the *sgr1* null lines and their antioxidant activities were investigated (Figure 4). In the *sgr1* #1-6 and *sgr1* #2-4 lines, *L**, representing lightness, did not show much difference in the breaker (Br) stage with the dark light green fruits, but it was observed that it decreased compared to the WT in the red Br+7 and Br+10 stages. In the *sgr1* null lines, *a**, measured according to the degree of redness of the tomato, had a negative value in the Br stage. Afterwards, in the Br+7 stage, in which the red color appears, the values of the *sgr1* #1-6 and *sgr1* #2-4 lines showed values of 13.3 to 14.2, respectively, and the WT with the pink fruit showed values of 2.5. In the Br+10 stage, in which the fruits are dark red, the *sgr1* #1-6 and *sgr1* #2-4 lines showed values of 22.3 to 23.8, respectively, and the WT, a mature pink tomato, showed values of 32.0. The values of *b** appeared similar among the *sgr1* null lines and WT.

### 2.5. Antioxidant Constituents

The content of secondary metabolites produced in the carotenoid biosynthesis pathway tended to gradually decrease as the fruit aged, except for lutein. However, the carotenoid content including lycopene and *β*-carotene increased in the *sgr1* null lines and WT. The content of *β*-carotene in the *sgr1* null lines decreased slightly compared to the WT in the Br stage, but more than doubled after the Br+7 stage. The lycopene contents of the *sgr1* null lines appeared to be twice as high as those of the WT in the Br stage, and the highest value was about 5.7 times in the Br+10 stage (Appendix A). The *α*-tocopherol content of the *sgr1* #1-6 and *sgr1* #2-4 lines was 61.5 and 59.7 μg/g, respectively, which was 4 times higher than that of the WT. However, the *γ*- and *δ*-tocopherol contents of the *sgr1* #1-6 and *sgr1* #2-4 lines showed similar results to the WT (Figure 5). The content of chlorophyll, a green photosynthetic pigment in plants derived from lipophilic extracts, was measured under the same LC conditions as the carotenoid analysis. The *sgr1* null lines showed significantly higher chlorophyll a (55.6 and 65.7 µg/g, respectively), compared to the wildtype (Appendix A). The free amino acid content including GABA was measured using 70% ethanol extract. The *sgr1* null lines showed significantly higher contents of L-glutamate, glutamine, aspartic acid, and asparagine than the wildtype. The average values of GABA were 11.8 mg/g and 12.3 mg/g in the *sgr1* #1-6 and *sgr1* #2-4 lines, respectively, which were relatively higher than the 9.7 mg/g in the wildtype. Vitamin C was measured as ascorbic acid equivalent (AAE) using 3% methyl acid extract. As a result, the vitamin C content of the *sgr1* #1-6 and *sgr1* #2-4 lines was 1.70 and 1.97 mg AAE/g, respectively, which was higher than that of the WT (Figure 6).

### 2.6. Antioxidant Activity of Lipophilic and Hydrophilic Extracts

The total phenol contents of the lipophilic extract did not show a significant difference between the *sgr1* null lines and WT. However, the total phenol contents of the hydrophilic extracts of the *sgr1* #1-6 and *sgr1* #2-4 lines were 27.2 ± 1.3 and 20.3 ± 0.2 µmol GAE/g, respectively, which were higher than those of the wildtype. In addition, the total flavonoid content of the hydrophilic extracts of the *sgr1* null lines were 10.8 µmol QE/g, which was higher than that of the wildtype. The antioxidant activity was measured by a DPPH radical scavenging analysis using lipophilic and hydrophilic extracts. In the *sgr1* #1-6 and *sgr1* #2-4 null lines, the DPPH radical scavenging activity of the hydrophilic extract was 1.2-3.2 times higher than that of the WT. Also, the antioxidant activity of the lipophilic and hydrophilic extracts of the *sgr1* #1-6 and *sgr1* #2-4 lines, with values of 27.3 ± 3.5 and 27.4 ± 1.8 (mol trio lox equivalents (TE)/g, respectively, was higher than that of the wildtype (Figure 7, Appendix A).

## 3. Discussion

Tomatoes are known to be native to South America and contain antioxidants such as lycopene, *β*-carotene, GABA, vitamins C and E, phenol, and flavonoids [2,18,19]. Carotenoids, natural lipophilic pigments, are essential pigments in photosynthetic organisms and are expressed as colors in flowers and fruits [20]. Many researchers have investigated the mechanism of carotenoid accumulation in tomato fruits by examining the expression level of biosynthetic genes present in the carotenoid biosynthetic circuit during the ripening process of the fruit [21,22,23]. Lycopene is a carotenoid hydrocarbon known in red tomatoes and has the effect of lowering the risk of chronic diseases and cardiovascular diseases [3]. Previously, various studies have been attempted to increase the lycopene content in tomato fruits. In our previous study, the levels of lycopene and *β*-carotene in mutant lines in which the *SISGR1* gene was knocked out using the CRISPR/Cas9 system were significantly higher than those in WT plants. Therefore, in this study, we investigated the growth characteristics, antioxidant content, and antioxidant activity of tomato skins of *sgr1* null lines ripened on vines. The results showed that the growth characteristics were almost similar in the *sgr1* null lines compared to WT, but the antioxidant activity was statistically significantly different (Appendix A). Typically, when tomatoes ripen from green to red, the brix, *β*-carotene, lycopene, and vitamin C contents are known to increase, while the GABA content decreases [18,24]. At the Br+10 stage, the *sgr1* #1-6 and *sgr1* #2-4 lines showed higher lycopene and *β*-carotene contents compared to the WT. Typically, lycopene accumulation in tomato fruit occurs during the ripening of the fruit as chlorophyll decomposes due to the programming of genes related to carotenoid biosynthesis, resulting in the loss of green color [25,26]. Additionally, compared with high-carotenoid-containing tomatoes from conventional breeding programs, the *sgr1* #1-6 and *sgr1* #2-4 lines resembled the *hyperpigmentation* (*hp-1*) mutant with increased lycopene and *β*-carotene contents [27]. The TSS and acidity of fruit are greatly influenced by the overall quality of sweetness, sourness, and flavor [28]. At the Br+10 stage, the total soluble solids and total yield of the *sgr1* null lines were higher than those of the WT (Table 1). Our current results are consistent with those of previously reported hyperpigmented variants [27,29] and suggest that robustness is negatively correlated with the TSS content and BAR. Meanwhile, the flesh color indicates the quality and freshness of tomato production. Especially when processing tomatoes, their value is graded according to their color [30]. In tomatoes, the red color is caused by the accumulation of lycopene [8,31], and we previously reported a high correlation between the lycopene content and the total carotenoid content [32]. The lycopene content of the *sgr1* null lines was 2-fold higher at the Br stage and approximately 5.7-fold higher at the Br+10 stage than in the WT (Appendix A). Normally, GABA accumulation in tomatoes takes place before the breaker stage and is rapidly catabolized thereafter [33]. Therefore, we measured the GABA content immediately after harvest in the pink stage of the *sgr1* #1-6 and *sgr1* #2-4 lines and the WT (Figure 6). Meanwhile, tomatoes are rich in ascorbic acid (vitamin C), which the body absorbs easily [22]. It has many health benefits, including defending against scurvy, preventing low-density lipoprotein oxidation, maintaining collagen, and improving neurodegenerative diseases [34,35]. The AAE content of the *sgr1* null lines was shown to be slightly higher than that of the WT (Figure 6). Phenols are a group of large molecules that act as natural antioxidants in plants [36]. They may prevent chronic diseases associated with excess free radicals by reducing oxidative stress [22,36]. The total phenols and flavonoid contents of the hydrophilic extracts of the *sgr1* #1-6 and *sgr1* #2-4 lines were higher than those of the WT. Flavonoids are the main components of total phenols [37]. Flavonoids have high antioxidant power, which contributes significantly to their health benefits [21]. Additionally, the antioxidant activity was investigated by a DPPH radical scavenging assay from lipophilic and hydrophilic extracts. In the *sgr1* null lines, the DPPH radical scavenging activity of the hydrophilic and lipophilic extracts was higher than that of the WT. Therefore, as the *SGR1* gene was knocked out, carotenoids such as lycopene, GABA, vitamin C, phenol, flavonoids, and the antioxidant capacity increased. In conclusion, the *sgr1* null lines have a higher total phenolic content, total flavonoid content, and DPPH radical scavenging activity of hydrophilic extracts compared to the WT, indicating that it exhibits high antioxidant activity.

## 4. Materials and Methods

### 4.1. Plant Materials

The *sgr1* null lines “*sgr1* #1-6 and *sgr1* #2-4”, previously reported by [9], were selected and grown in the GMO greenhouse in the Horticulture Department of Horticultural Biotechnology in Hankyoung National University (Korea) in spring 2023. The fertilizer solution was used after adjustment to pH 5.5–5.8 and EC 2.0–2.2 dS m-1 at each growth stage. Harvest maturation was achieved 72 days after transplantation; fruits were harvested from the third cluster of each plant, and fruits at the edge of the cluster were not used in the experiment. In addition, a total of five stages were divided into MG, Br, Br+4, Br+7, and Br+10 for each stage of development of tomato fruit. The non-physical defects and uniform size of the fruit were measured in the pink phase using the USDA Tomato Aged Color Classification Chart [38]. Classification of the pink phase was carried out again in the laboratory using color tables reported by USDA. After selection, physicochemical data were immediately collected and samples for *β*-carotene, lycopene, ascorbic acid, flavonoids, polyphenols, and antioxidant activity assays were frozen by liquid nitrogen and stored in a freezer (−80 °C) for approximately 3 weeks until the assay was performed. Samples for GABA assays were freeze-dried, ground into fine powders, filtered through a 40 μm mesh, and then stored at −20 °C until extraction.

### 4.2. Expression Analysis of SISGR1 Gene

RNA was extracted from roots, stems, leaves, and fruits from Br stage, and then analyzed for expression of the *SISGR1* gene using qRT-PCR. Primers used in this assay are listed in Appendix A.

### 4.3. Analysis of SGR1 Protein in sgr1 Null Lines

Western blot analysis was conducted according to the method reported by Jiang et al. [39]. Total soluble proteins extracted from ripened fruit tissues of the *sgr1* null lines (*sgr1* #1-6 and *sgr1* #2-4) were separated by 15% SDS-PAGE in 100 V of Tris–glycine buffer (25 mM Tris (pH8.5), 200 mM glycine). The separated protein bands were transferred from gel to Hybond membranes (Roche, Basel, Switzerland) using Minitrans Blot Electrophoresis Transfer Cells (BioRad, Hercules, CA, USA) in transfer buffer including 40 mM glycine, 50 mM Tris (pH 8.3), 0.04% SDS, and 20% methanol for 2 h. The non-specific antibody reaction was blocked by incubating the membrane in 25 mL of 5% fat-free formula (TBS with 0.05% Tween-20) in TBST buffer with gentle stirring at room temperature for 12 h. The membranes were incubated in TBST buffer at a 1:3000 dilution with rabbit anti-*SGR1* IgG, stirred gently for 2 h, and washed three times with TBST buffer. The membranes were then incubated in a 1:5000 dilution of horseradish peroxidase-conjugated anti-rabbit IgG (TakaRa, Kusatsu, Japan) in TBST buffer for 2 h with gentle shaking and washing three times in TBST buffer and once in TMN buffer. They were then banded using TMB substrates (Roche) in TMN buffer including 5 mM MgCl2, 100 mM Tris (pH 9.5), and 100 mM NaCl.

### 4.4. Total Soluble Solids, Total Acids of Pulp, and Brix Acid Ratio

The total solute solids (TSS) content was measured according to the protocol of [18], previously reported using fresh fruit samples at room temperature. The TA content was measured as citric acid equivalent (μg CAE/g) by diluting each sample extracted in juice form with distilled water and filtering it with a 0.45 μm filter. The pH value was adjusted to 8.1 and calculated as the citric acid equivalent (μg CAE/g). BAR was calculated by dividing the TSS value by the TA value.

### 4.5. Analysis of Carotenoids and Tocopherols

Analysis of carotenoids and tocopherol was performed after separation of lipophilic extracts from tomato pulp using acetone (0.01% butylated hydroxytoluene (BHT)) according to a previously reported method [40,41]. HPLC analysis was conducted using an Agilent 1260 HPLC system (Hewlett-Packard, Waldbronn, Germany), and the gradient system of the eluent was slightly modified from the previously reported one. HPLC-grade water, methanol, and methyl-tert-butyl ether (MTBE) were used as solvents. The solvent mixtures were prepared in a ratio of methanol–MTBE–water (81:15:4; A) and methanol–MTBE–water (6:90:4; B). The chromatogram was accommodated by gradient elution under the following conditions: 0 to 15 min, 0% B; 15 to 50 min, 100 min B; 50 to 60 min, and 100 percent B. The flow rate was maintained at 0.7 mL/min and the temperature was 30 °C. Carotenoid standards, such as all-trans-*β*-carotene, lutein, and all-trans-lycopene from Carotenature (Münsingen, Switzerland) were analyzed for peaks at 450 nm absorbance. The same LC conditions were used to quantify chlorophyll *a* and *b* (Sigma-Aldrich). Under these conditions, each standard peak was eluted from the following tR (min): violaxanthin 11.1, chlorophyll b 18.8, lutein 20.9, zeaxanthin 25.8, chlorophyll a 26.8, 13 Z-*β*-carotene 35.7, α-carotene 36.5, all-trans-*β*-carotene 38.4, 9 Z-*β*-carotene 39.6, all-trans-lycopene 52.6, and 5 Z-lycopene 53.3. Tocopherol analysis was performed with slight modifications to the method previously reported [34]. The same HPLC system as used in the carotenoid analysis was used with the same eluent. The chromatogram was performed by gradient elution under the following conditions: 0 to 16 min, 0% B; 16 to 20 min, 100% B; 20 to 30 min, and 100% B. The flow rate was maintained at 0.7 mL/min and at 30 °C. Peak detection was analyzed by fluorescence (Ex. 298 nm and Em. 325 nm). Under these conditions, the standard tocopherol peaks (Sigma-Aldrich, St. Louis, MO, USA) were eluted at the following tR (min): *δ*-tocopherol 10.1, *γ*-tocopherol 11.5, *β*-tocopherol 12.2, and *α*-tocopherol 13.3.

### 4.6. Analysis of GABA and Free Amino Acids

The content of free amino acids including GABA was measured according to a previously reported method [3]. A total of 250 mg of lyophilized tomato was extracted by ultrasonic treatment with 10 mL of 70% ethanol three times, each for 10 min. The extract was centrifuged at 5700× *g* for 10 min at room temperature, and the upper layer was filtered through a 0.22 μm filter. The obtained filtrate was measured with a Bio-chrom 30+ amino acid analyzer system (Biochrom Ltd., Cambridge, UK) using a Lithium Accelerated Resin H-1649 column (Biochrom Ltd., Cambridge, UK) and absorbance was measured at 570 nm.

### 4.7. Vitamin C Content

The vitamin C content was measured according to the previously reported method [31]. The 250 mg of lyophilized tomato was sonicated three times, each for 10 min, and extracted with 3% methic acid. The extract was centrifuged at 5700× *g* and 4 °C for 10 min, and the obtained supernatant was filtered through a 0.45 μm filter. HPLC analysis was conducted using 0.2 M KH2PO4 (pH 2.2, solvent A) and methanol (solvent B) on an Agilent 1260 HPLC system and a YMC ODS C18 column (4.6 × 250 mm, 4 μm). Gradient elution conditions were as follows: 0 to 5 min, 0% B; 15 to 23 min, 100% B; 23 to 24 min, 0% B; 24 to 25 min, and 0% B. A volume of 20 μL was injected at a flow rate of 0.7 mL/min. The column temperature was 30 °C and the absorbance was measured at 254 nm.

### 4.8. Total Phenolic and Total Flavonoid Contents

In order to measure the total phenol content and flavonoid content, a lipophilic extract and a hydrophilic extract were used. The hydrophilic extract was prepared by ultrasonicating 250 mg of freeze-dried tomatoes and 10 mL of methanol three times, respectively, for 10 min. The extract was centrifuged at 5700× *g* at 4 °C for 10 min, and the obtained supernatant was filtered through a 0.45 μm filter. The total phenolic content was measured by Folin–Ciocalteu analysis [42]. After adding 500 μL of 10% Folin–Ciocalteu phenol reagent to 100 μL of each extract, 400 μL of sodium carbonate was added, allowed to react at room temperature for 10 min, and then centrifuged at 3000× *g* for 5 min. After transferring 200 μL of the supernatant to 96 wells, the absorbance was measured at 765 nm. A calibration curve was obtained with gallic acid, and the phenol content was expressed as gallic acid equivalent (μmol GAE/g) per gram. Total flavonoid content was measured by diethylene glycol analysis [42]. To 20 μL of each extract, 10 μL of 4 M NaOH and 170 μL of 90% diethylene glycol were added and allowed to react at room temperature. After 10 min, absorbance was measured at 420 nm. Calibration curves were obtained with quercetin, and flavonoid content was expressed as quercetin equivalents (μmol QE/g) per gram.

### 4.9. Antioxidant Activity Test

DPPH radical scavenging activity was analyzed by a previously reported method [43]. A volume of 20 μL of each sample was added to 180 μL of a 0.2 mM 1,1-diphenyl-2-picrylhydroxyl (DPPH) solution in each tube. After 10 min, absorbance was measured at 520 nm. Antioxidant activity was expressed in terms of Trolox (Sigma-Aldrich, St. Louis, MO, USA) gram-equivalent (μmol TE/g).

### 4.10. Statistical Analysis

Statistical analysis was conducted using RStudio 4.3.1 software (RStudio, Boston, MA, USA). Repeated measures analysis of variance compared the mean scores across the quantitative categories. Post hoc pairwise comparisons were conducted using paired *t*-tests with the Bonferroni correction.

## Figures and Tables

**Figure 1 ijms-25-05111-f001:**
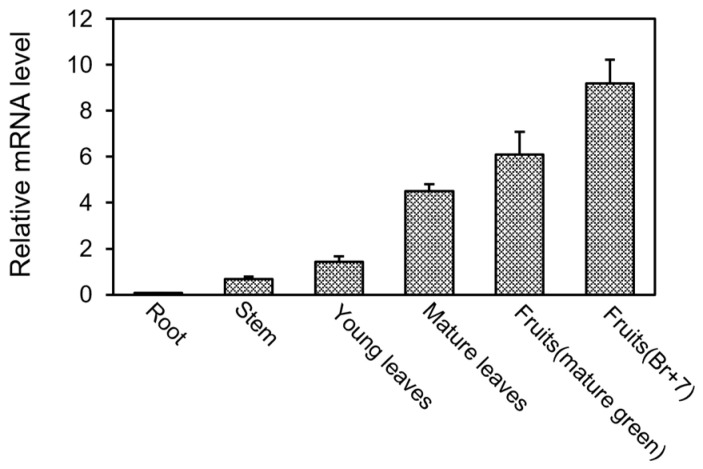
Expression pattern of *SGR1* gene in each organ of tomato. The relative mRNA level of the *SGR1* gene was measured in plant organs such as roots, stems, leaves, and fruits.

**Figure 2 ijms-25-05111-f002:**
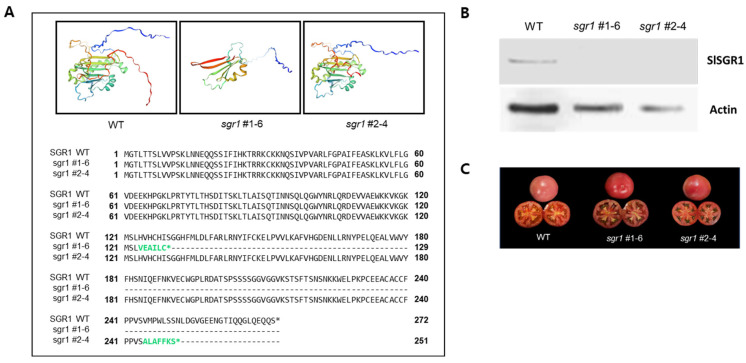
Protein structure and expression analysis of *sgr1* null lines. (**A**) Protein structure analysis of *sgr1* null lines. Green letters indicate protein sequences translated by frameshifting and premature termination. Asterisks indicate stop codon. (**B**) Western blotting analysis for expression levels of SGR1. (**C**) Phenotype of fruit (Br+7) obtained from the *sgr1* null lines.

**Figure 3 ijms-25-05111-f003:**
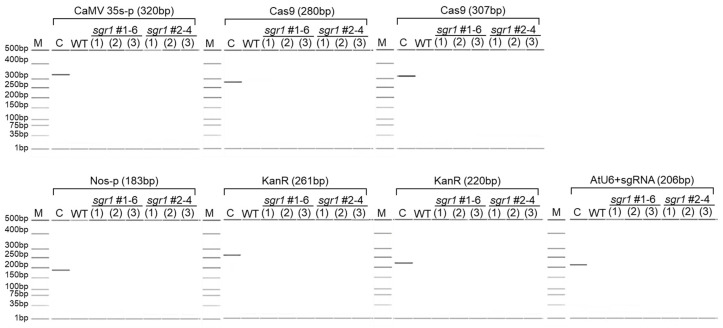
Electrophoretic patterns after PCR analysis of *sgr1* null lines. Three plants per mutant line were used for analysis. M; 500 bp marker, C; PCR product amplified from pKAtC vector.

**Figure 4 ijms-25-05111-f004:**
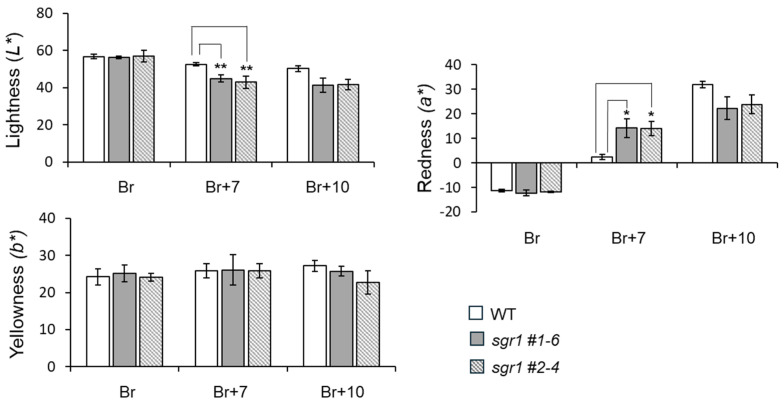
Colorimetric evaluation of *sgr1* null lines at the ripening development stage. The bars show a significant difference (* 0.01 < *p* < 0.05, ** 0.001 < *p* < 0.01) between WT and *sgr1* null lines (Bonferroni’s test). Vertical bars show the standard error of the mean for three replicates.

**Figure 5 ijms-25-05111-f005:**
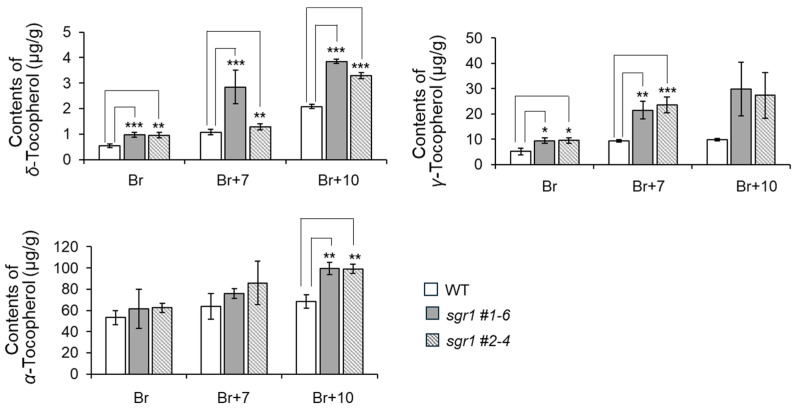
Tocopherol contents of WT and *sgr1* null lines at the ripening development stage. The bars show a significant difference (* 0.01 < *p* < 0.05, ** 0.001 < *p* < 0.01, *** *p* < 0.001) between WT and *sgr1* null lines (Bonferroni’s test). Vertical bars show the standard error of the mean for three replicates.

**Figure 6 ijms-25-05111-f006:**
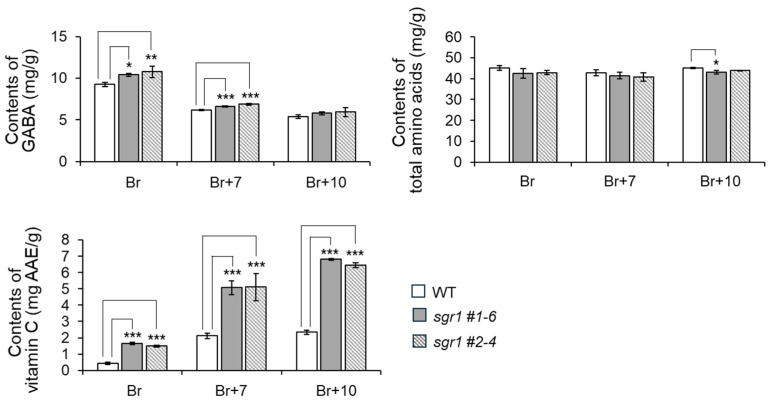
Free amino acid contents of WT and *sgr1* null lines at the ripening development stage. The bars show a significant difference (* 0.01 < *p* < 0.05, ** 0.001 < *p* < 0.01, *** *p* < 0.001) between WT and *sgr1* null lines (Bonferroni’s test). Vertical bars show the standard error of the mean for three replicates.

**Figure 7 ijms-25-05111-f007:**
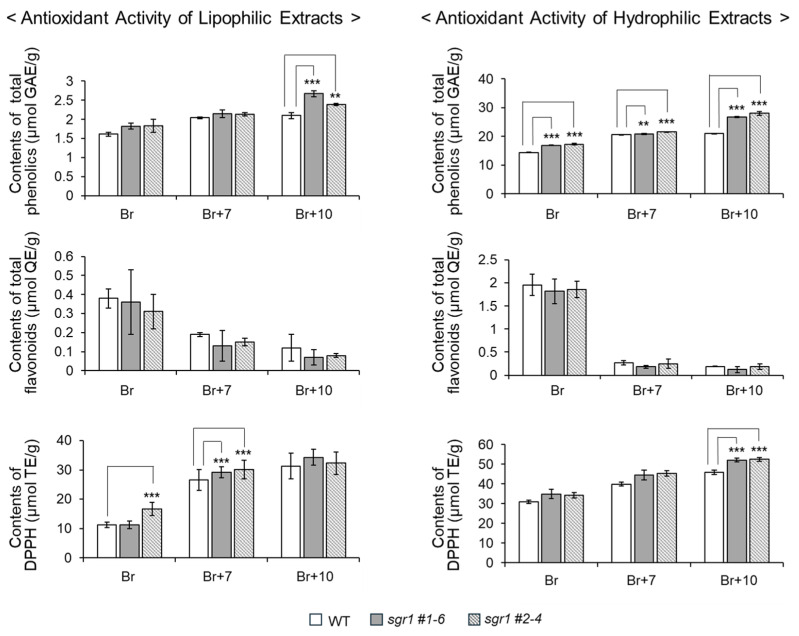
Total phenol contents of WT and *sgr1* null lines at the ripening development stage. The bars show a significant difference (** 0.001 < *p* < 0.01, *** *p* < 0.001) between WT and *sgr1* null lines (Bonferroni’s test). Vertical bars show the standard error of the mean for three replicates.

**Table 1 ijms-25-05111-t001:** Firmness, TSS, TA, and BAR of WT and *sgr1* null lines at red ripe stage.

Lines	Firmness (N)	TSS (°Brix)	TA (mg CAE/10 g)	BAR
WT	2.80 ± 0.21	4.70 ± 0.19	0.49 ± 0.03	9.47 ± 0.25
*sgr1* #1-6	2.03 ± 0.24 *	5.79 ± 0.14 ***	0.41 ± 0.04 *	14.10 ± 1.28 ***
*sgr1* #2-4	2.13 ± 0.22	5.71 ± 0.15 ***	0.42 ± 0.01	13.60 ± 0.22 ***

Data represent the means ± SD of three replicate experiments. Asterisks show significant differences (* 0.01 < *p* < 0.05, *** *p* < 0.001) between *sgr1* null lines and WT (Bonferroni’s test).

## Data Availability

The original contributions presented in the study are included in the article/Appendix A, further inquiries can be directed to the corresponding authors.

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
