# Peer review of "Physicochemical Properties and Antioxidant Activity of CRISPR/Cas9-Edited Tomato SGR1 Knockout (KO) Line"

_ijms, 2024, doi:10.3390/ijms25105111_

Round 1
Reviewer 1 Report
Comments and Suggestions for Authors
Title:
Physicochemical Properties and Antioxidant Activity of Tomato SGR1-Knockout (KO) Lines Using CRISPR/Cas9 System
Authors: Jin Young Kim, Dong Hyun Kim, Me-Sun Kim, Yu Jin Jung, and Kwon Kyoo Kang
Submitted in: IJMS
While the study presents an interesting approach to enhancing the nutritional quality of tomatoes through CRISPR/Cas9 mediated knockout of the SGR1 gene, several major concerns need to be addressed before the manuscript can be considered for publication.
The recommendation is for rejection.
Major Concerns:
- Lack of Experimental Controls:
- The study's design significantly lacks appropriate controls, especially concerning environmental and genetic background variability. For the CRISPR/Cas9 experiments, it is crucial to include both non-edited plants and plants edited without targeting any gene (mock controls) to differentiate between the effects of the CRISPR/Cas9 system itself and the specific gene knockout. This oversight questions the validity of attributing observed phenotypic differences solely to the SGR1 knockout.
- Inadequate Statistical Analysis:
- The manuscript does not provide sufficient details on the statistical methods used for data analysis, and there seems to be a lack of rigorous statistical treatment to support the conclusions drawn. The variability within the experimental groups is not adequately addressed, and no information on replicates, randomization, or statistical tests is provided, which are critical for validating the study's findings.
- Insufficient Characterization of Mutant Lines:
- While the study aims to investigate the effects of the SGR1 knockout on tomato fruits' physicochemical properties and antioxidant activity, there is an apparent lack of molecular characterization of the mutant lines. Details on the efficiency of the CRISPR/Cas9 system, off-target effects, and confirmation of the knockout at the DNA and protein levels are scant. Such characterizations are fundamental to ensuring that the observed phenotypes are indeed a result of the SGR1 knockout.
- Limited Scope of Antioxidant Activity Assessment:
- The manuscript focuses narrowly on a select few antioxidant components and activities, which does not fully capture the complex nature of antioxidant properties in tomatoes. Expanding the range of antioxidants analyzed and including assays that measure overall antioxidant capacity would provide a more comprehensive understanding of the CRISPR/Cas9 system's impact on tomato nutritional quality.
- Overgeneralization of Results:
- The discussion section tends to overgeneralize the study's findings without adequately considering the specific genetic, developmental, and environmental contexts that significantly influence phenotypic outcomes in genetically modified organisms. The implications of the study for tomato breeding and nutritional improvement are discussed without sufficient caution or acknowledgment of the complexities involved.
Recommendations for future improvement of your submission
- Addressing Experimental Controls: The authors must include appropriate control groups and detail their use in the manuscript to strengthen the study's validity.
- Enhancing Statistical Rigor: A comprehensive description of the statistical analysis, including the tests used, the number of replicates, and how variability was handled, should be included. Statistical significance should be clearly stated for all comparisons.
- Molecular Characterization of Mutant Lines: The authors should provide detailed molecular characterizations of the mutant lines, including evidence of the specific knockout at the DNA and protein levels and assessments of potential off-target effects.
- Broadening Antioxidant Analysis: Expand the range of antioxidant components and activities analyzed to provide a more holistic view of the nutritional implications of SGR1 knockout.
- Cautious Discussion and Generalization: The discussion should be done to more carefully reflect the limitations of the study and avoid overgeneralization of the findings. The potential implications for tomato breeding and nutritional improvement should be discussed with appropriate caution.
Here are some detailed remarks to guide the authors in further developing the manuscript:
1. Inadequate Description of CRISPR/Cas9 Methodology
Remark: The manuscript provides an insufficient description of the CRISPR/Cas9 editing process, including the design of guide RNAs, the specificity of target sites, and the method used to deliver the CRISPR/Cas9 components into the tomato plant cells. For reproducibility and to assess the risk of off-target effects, it is essential to include detailed information about the gRNA sequences used, the Cas9 variant employed, and the method of transformation (e.g., Agrobacterium-mediated transformation, biolistic delivery). Additionally, a more comprehensive explanation of the selection process for edited plants and the criteria for confirming SGR1 knockout at the genomic level would significantly enhance the manuscript's methodological rigor and its importance.
2. Lack of Detailed Phenotypic Characterization
Remark: The manuscript briefly mentions increases in certain antioxidants and improvements in physicochemical properties without providing a thorough phenotypic characterization of the SGR1-KO lines compared to wild-type controls. Detailed phenotypic analysis, including morphological characteristics of the plants (e.g., plant height, leaf size, fruit set), as well as a quantification of fruit yield and quality parameters (e.g., fruit size, weight, texture), is critical for understanding the broader agricultural implications of SGR1 knockout. Moreover, data on the developmental stages of fruits analyzed and environmental conditions during plant growth are crucial for contextualizing the results.
3. Superficial Statistical Analysis
Remark: The manuscript's statistical treatment appears superficial, with a lack of detailed statistical analysis that would allow for the robust interpretation of the data. Specifically, the use of appropriate statistical tests, justification for their selection, and details on the handling of replicate samples, variability, and potential outliers are conspicuously absent. For each experimental outcome, the manuscript should present a comprehensive statistical analysis, including assumptions checked, exact p-values, effect sizes, and confidence intervals where applicable, to substantiate the claims made.
4. Overlooked Environmental and Genetic Background Effects
Remark: The study fails to address the potential effects of environmental variables and genetic background on the observed phenotypic changes in SGR1-KO lines. Given the significant impact of these factors on gene expression and plant metabolism, it is imperative to conduct experiments under controlled conditions, with detailed reporting of environmental parameters, and to consider the genetic background of the tomato lines used. Without this information, it is difficult to ascertain whether the observed changes are solely attributable to the SGR1 knockout or if they might be influenced by uncontrolled variables.
5. Insufficient Discussion of Potential Off-Target Effects
Remark: While the manuscript mentions the targeted knockout of the SGR1 gene, there is an evident lack of discussion regarding the assessment and implications of potential off-target effects of the CRISPR/Cas9 system. It is critical to perform genome-wide analyses to identify and quantify any unintended edits that may have occurred, as these could have unforeseen impacts on the plant's phenotype, fitness, and nutritional content. The manuscript would greatly benefit from including data on off-target analysis methodologies used (e.g., whole-genome sequencing, targeted deep sequencing of potential off-target sites), results of these analyses, and how any identified off-target edits were accounted for in the interpretation of the study's outcomes.
Comments on the Quality of English LanguageAverage quality
Author Response
Response of Reviewer1
While the study presents an interesting approach to enhancing the nutritional quality of tomatoes through CRISPR/Cas9 mediated knockout of the SGR1 gene, several major concerns need to be addressed before the manuscript can be considered for publication.
Major Concerns:
- Lack of Experimental Controls:
The study's design significantly lacks appropriate controls, especially concerning environmental and genetic background variability. For the CRISPR/Cas9 experiments, it is crucial to include both non-edited plants and plants edited without targeting any gene (mock controls) to differentiate between the effects of the CRISPR/Cas9 system itself and the specific gene knockout. This oversight questions the validity of attributing observed phenotypic differences solely to the SGR1 knockout.
--- Thank you for reviewing from various angles. The CRISPR/Cas9 experiment pointed out by the reviewer is one we previously published, IJMS 2023, 24,109. https://www.mdpi.com/1422-0067/24/1/109. The above paper describes the generation and characteristics of the sgr1 #1−6 and sgr1 #2−4 lines. Reviewer 1, please click on the published paper.
- Inadequate Statistical Analysis:
The manuscript does not provide sufficient details on the statistical methods used for data analysis, and there seems to be a lack of rigorous statistical treatment to support the conclusions drawn. The variability within the experimental groups is not adequately addressed, and no information on replicates, randomization, or statistical tests is provided, which are critical for validating the study's findings.
--- Thank you. The statistical analysis method pointed out in Review 1 is pointed out in Materials and Methods, and the significance test using Bonferroni's t-test and three replicate experiments are indicated at the bottom of the presented tables and figures.
- Insufficient Characterization of Mutant Lines:
While the study aims to investigate the effects of the SGR1 knockout on tomato fruits' physicochemical properties and antioxidant activity, there is an apparent lack of molecular characterization of the mutant lines. Details on the efficiency of the CRISPR/Cas9 system, off-target effects, and confirmation of the knockout at the DNA and protein levels are scant. Such characterizations are fundamental to ensuring that the observed phenotypes are indeed a result of the SGR1 knockout.
--- Thank you. The molecular properties of the sgr1 mutant series, as noted by reviewer 1, were well characterized in a previous report (https://www.mdpi.com/1422-0067/24/1/109). However, in our previous paper we looked at knockout information and phenotypic changes at the DNA level. Therefore, in this paper, the protein was not expressed using Western blot analysis using sgr1 #1-6 and sgr1 #2-4 lines in which the gene was knocked out. Therefore, we have performed all the basic demonstrations of gene knockout pointed out in Review 1.
- Limited Scope of Antioxidant Activity Assessment: The manuscript focuses narrowly on a select few antioxidant components and activities, which does not fully capture the complex nature of antioxidant properties in tomatoes. Expanding the range of antioxidants analyzed and including assays that measure overall antioxidant capacity would provide a more comprehensive understanding of the CRISPR/Cas9 system's impact on tomato nutritional quality.
--- Thank you. This manuscript is to discuss the Physicochemical Properties and Antioxidant Activity between sgr1 #1-6 and sgr1 #2-4 lines fruits generated by CRISPR/Cas9 and control fruits. For the accuracy of the experiment, sgr1 #1-6 and sgr1 #2-4 lines and control lines were grown under the same conditions. For all analyses, the third bunch of each sample was harvested, and fruits at the edge of the cluster were not used in the experiments. Additionally, it did not extend the range of antioxidants for tomato nutritional quality suggested by reviewer 1. We limited ourselves to the Physicochemical Properties and Antioxidant Activity of the sgr1 #1-6 and sgr1 #2-4 lines.
- Overgeneralization of Results:
The discussion section tends to overgeneralize the study's findings without adequately considering the specific genetic, developmental, and environmental contexts that significantly influence phenotypic outcomes in genetically modified organisms. The implications of the study for tomato breeding and nutritional improvement are discussed without sufficient caution or acknowledgment of the complexities involved.
--- Thank you. I put a table in Supplementary 3 that measured the phenotype of the sgr1 #1-6 and sgr1 #2-4 lines grown by the authors. And I explained the description in lane 199-206 as follows. In addition, the molecular properties were previously submitted and sufficiently explained in the crab limitation paper. For the tomato development and environmental considerations pointed out by Reviewer 1, the sgr1 #1-6 and sgr1 #2-4 strains and controls were used under the same cultivation conditions and simultaneously with the third fruit as described previously. Therefore, it is thought that the strains obtained in this study could be used as ingredients for tomato breeding.
In our previous study, the levels of lycopene and β-carotene in mutant lines in which the SISGR1 gene was knockout using the CRISPR/Cas9 system were significantly higher than those in WT plants. Therefore, in this study, we investigated the growth characteristics, antioxidant content and antioxidant activity of tomato skins ripened on vines of sgr1 mu-tant lines. As a result, compared to WT, the growth characteristics were almost similar in the sgr1 #1-6 and sgr1 #2-4, but the antioxidant activity was statistically significant (Supplementary Table 3).
Recommendations for future improvement of your submission
-- Thank you The above points pointed out by reviewer 1 are well explained in Major Concerns.
Reviewer 2 Report
Comments and Suggestions for Authors
This is an interesting work, which should be published in IJMS. But there are some major comments need to be addressed by the authors.
1. There are too many abbreviations in this paper, please check them carefully. The full name should be given and only given when they firstly appeared, such as WT and DPPH in abstract, “TSS (total soluble solids)” in introduction, “The total soluble solids (TSS)” in results and “The total solute solids (TSS)” in methods.
2. Some sentences are hard to read. e.g. “In addition, the flavor intensity, the sgr1 mutant lines 18 showed higher ascorbic acid content and carotenoid (lycopene and β-carotene) accumulation com-19 pared to the WT.”
3. “A functional compound 13 that promotes health, γ-aminobutyric acid (GABA), has also been shown to accumulate in high 14 amounts in tomato fruits.” This sentence seems to indicate GABA was especially important for this study, but it is not.
Replace it with a sentence to introduce sgr1gene or gene editing lines may be more appropriate.
4. “Physicochemical Properties and Antioxidant Activity of CRISPR/Cas9 Edited Tomato SGR1-Knockout Lines” may be more suitable.
5. The discussions should be more logical and highlight some key points.
6. Only DPPH clearing test was weak for the evaluation of antioxidant activities. Why do not consider Fe reduction, ROS clearing test, and some others? They are easy to preform.
Author Response
Response od reviewer 2
- There are too many abbreviations in this paper, please check them carefully. The full name should be given and only given when they firstly appeared, such as WT and DPPH in abstract, “TSS (total soluble solids)” in introduction, “The total soluble solids (TSS)” in results and “The total solute solids (TSS)” in methods.
--- Thank you. We have modified as pointed out by reviewer 2.
- Some sentences are hard to read. e.g. “In addition, the flavor intensity, the sgr1 mutant lines 18 showed higher ascorbic acid content and carotenoid (lycopene and β-carotene) accumulation com-19 pared to the WT.”
----- Thank you, I've revised it as follows, referring to what reviewer 2 pointed out.
Additionally, the sgr1 mutant line accumulated higher levels of flavor-inducing ascorbic acid and carotenoids (lycopene and β-carotene) compared to WT.
- “A functional compound 13 that promotes health, γ-aminobutyric acid (GABA), has also been shown to accumulate in high 14 amounts in tomato fruits.” This sentence seems to indicate GABA was especially important for this study, but it is not.
Replace it with a sentence to introduce sgr1gene or gene editing lines may be more appropriate.
--- Thank you, I've revised it as follows, referring to what reviewer 2 pointed out.
SlSGR1 encodes a STAY-GREEN protein that plays a critical role in the regulation of chlorophyll degradation in tomato leaves and fruits.
- “Physicochemical Properties and Antioxidant Activity of CRISPR/Cas9 Edited Tomato SGR1-Knockout Lines” may be more suitable.
---Thank you. We have accepted and modified the points pointed out by reviewer 2.
- The discussions should be more logical and highlight some key points.
---- Thank you, we have the following emphasis at the end of Discussion, referring to what you pointed out in reviewer 2.
In conclusion, the sgr1 #1-6 and sgr1 #2-4 lines, which are knockout lines of tomato SGR genes, have a higher content of lycopene and carotenoids compared to WT, indicating high antioxidant activity.
- Only DPPH clearing test was weak for the evaluation of antioxidant activities. Why do not consider Fe reduction, ROS clearing test, and some others? They are easy to perform.
--- Thank you. As reviewer 2 knows well, we chose an easy one to evaluate antioxidant activity.
Round 2
Reviewer 1 Report
Comments and Suggestions for Authors
All mistakes were corrected
Comments on the Quality of English Languageaverage quality
Author Response
Thank you for reviewing our manuscript.
Reviewer 2 Report
Comments and Suggestions for Authors
There are still some core comments to this submission.
Major
1. Figure 2A in this submission was the same as previously published Figure 1A in reference [9]! Figure 2B in this submission is part of Figure 2B in reference [9]! Figure 4A in this submission VS Figure 1B in reference [9]!
2. Some data in figure 6 has also been published in Table 3 and Table 4 in reference [9]!
3. As the generation of "sgr1 #1-6 and sgr1 #2-4" has been published in reference [9], the related description should only restricted in Materials, but not Results. “2.2. Generation of sgr1 homozygous mutation lines” should be deleted or changed to an another statement only with new data and new results.
4. Based on the above information, I also doubt the originality of Figure 1. Dose the expression pattern of SGR1 in tomato tissues have been published in elsewhere.
Minor
1. The full name of the abbreviation is only given when it first appears.
2. The format of the figures needs to be consistent. Subfigures with A, B, C…. The significance markers are also inconsistent.
Author Response
Response of review 2
Major
- Figure 2A in this submission was the same as previously published Figure 1A in reference [9]! Figure 2B in this submission is part of Figure 2B in reference [9]! Figure 4A in this submission VS Figure 1B in reference [9].
--- Thank you. Based on what the review 2 pointed out, the authors set Figure 2 as Supplementary Figure S1.
- Some data in figure 6 has also been published in Table 3 and Table 4 in reference [9].
--- Thank you. Based on what the review 2 pointed out, the authors set Figure 6 A and Figure 6 C as Supplementary Figure S2A and Figure S2B.
- As the generation of "sgr1 #1-6 and sgr1 #2-4" has been published in reference [9], the related description should only restricted in Materials, but not Results. “2.2. Generation of sgr1 homozygous mutation lines” should be deleted or changed to an another statement only with new data and new results.
--- Thank you. The "sgr1 #1-6 and sgr1 #2-4" lines pointed out in Review 2 are null lines, which are 2-3 generations advanced by selfing the lines presented in the reference [9] published by the authors.
Therefore, generation was removed and expressed as null lines.
- Based on the above information, I also doubt the originality of Figure 1. Dose the expression pattern of SGR1 in tomato tissues have been published in elsewhere.
--- Thank you. The expression analysis of the SGR1 gene pointed out by reviewer 2 has been published elsewhere. However, the authors think that it is original by examining the expression level when it is fruits (mature green) and fruits (Br+7).
Minor
- The full name of the abbreviation is only given when it first appears
--- We have revised all the contents pointed out by reviewer 2.
- The format of the figures needs to be consistent. Subfigures with A, B, C…. The significance markers are also inconsistent.
--- We have revised all the contents pointed out by reviewer 2.
Round 3
Reviewer 2 Report
Comments and Suggestions for Authors
Thank the authors about their effort to improve the manuscript. It should be published after some minor revisions.
1. It should clearly indicate that the figure has been published in reference [9] in the legend of Supplementary Figure S1.
2. Line 42, there should no spaces for % and °C.
3. The abbreviations are still not standardized.
Abstract is not belonging to the main text. Thus, full name should be given when it firstly appeared in abstract and main text, respectively. e.g. the full name of TSS should be given at Line17 and Line 72, but not Line 277. Please carefully check the others.
Author Response
Response of Review 2
- It should clearly indicate that the figure has been published in reference [9] in the legend of Supplementary Figure S1.
- Thank you. We inserted that Published in reference from [9], International Journal of Molecular Sciences in the legend of Supplementary Figure S1
- Line 42, there should no spaces for % and °C.
- Thank you. We fixed it all. - The abbreviations are still not standardized.
- Thank you. We fixed it all.